# Comparative Metagenomics of Anaerobic Digester Communities Reveals Sulfidogenic and Methanogenic Microbial Subgroups in Conventional and Plug Flow Residential Septic Tank Systems

**James Naphtali, Alexander W. Y. Chan, Faizan Saleem, Enze Li, Jacob Devries and Herb E. Schellhorn ***

Department of Biology, McMaster University, 1280 Main Street West, Hamilton, ON L8S 4K1, Canada; naphtaj@mcmaster.ca (J.N.); chanw19@mcmaster.ca (A.W.Y.C.); saleef4@mcmaster.ca (F.S.); lie40@mcmaster.ca (E.L.); devrij7@mcmaster.ca (J.D.)
* Correspondence: schell@mcmaster.ca; Tel.: +1-905-525-9140 (ext. 27316)

**Abstract:** On-site wastewater treatment systems (OWTS) are primarily monitored using physiochemical factors, including chemical oxygen demand (COD) and residual total suspended solids (TSS), which are indirect measures of the microbial action during the anaerobic digestion process. Changes in anaerobic digester microbial communities can alter the digester performance, but this information cannot be directly obtained from traditional physicochemical indicators. The potential of metagenomic DNA sequencing as a tool for taxonomic and functional profiling of microbial communities was examined in both common conventional and plug flow-type anaerobic digesters (single-pass and recirculating). Compared to conventional digesters, plug flow-type digesters had higher relative levels of sulfate-reducing bacteria (*Desulfovibrio* spp.) and hydrogenotrophic methanogens (*Methanospirillum* spp.). In contrast, recirculating anaerobic digesters were enriched with denitrifier bacteria and hydrogenotrophic methanogens, and both were significantly correlated with physicochemical factors such as COD and TSS. Stratification of microbial communities was observed along the digester treatment process according to hydrolytic, acidogenic, acetogenic, and methanogenic subgroups. These results indicate that the high-throughput DNA sequencing may be useful as a monitoring tool to characterize the changes in bacterial communities and the functional profile due to differences in digester design in on-site systems.

**Keywords:** anaerobic digestion; methanogenesis; septic tank; whole-metagenome sequencing; metagenomics; bacterial communities



## 1. Introduction

Human and animal sewage wastes containing nitrogenous and phosphorous inorganic compounds such as ammonia, nitrate, and phosphate, along with organic compounds, directly contribute to increased nutrient loading in water bodies, causing eutrophication [1]. On-site anaerobic septic tank systems are a primary means for treating wastewater by reducing input waste to residual solids due to the combined metabolic activities of complex microbial communities in anaerobic digester systems [2,3]. The metabolic pathways primarily expressed by hydrolytic, acidogenic, acetogenic, and methanogenic bacterial communities drive anaerobic biomass degradation [4].

Anaerobic digestion operational efficiency can be measured using physicochemical parameters, including chemical oxygen demand (COD), total residual solids (TSS), and biogas production [5]. Observed differences in these parameters are indicators of digester performance, which, in turn, are affected by the composition of the complex digester microbial communities [5]. Influent pre-treatment methods such as microwaves and static magnetic fields can alter anaerobic digester microbial communities [6,7], and such alterations in response to pre-treatment methods and physicochemical parameters can result in

enhanced biogas production and optimal anaerobic digestion [8–10]. Therefore, anaerobic digester design and operation parameters such as hydraulic retention times (HRT) and influent pre-treatment methods must be adjusted to achieve optimal functionality [11]. Though characterization of microbial communities and their metabolic potential is likely important for optimization of the anaerobic digestion process, there is little information available regarding changes in the microbial community (microbiome) in anaerobic digesters. Such information could be useful for optimizing digester performance or tracking digester failure [12,13].

Some microbial functions with respect to anaerobic digestion can be specifically enhanced by optimizing physicochemical factors. For example, a change in pH toward the acidic side can enhance the microbial hydrolysis potential [14]. However, microbial community composition analysis can be a better indicator of digester function than physicochemical parameters due to the high robustness, resilience, and ecological flexibility of the microbiome [15].

Anaerobic digester communities such as methanogens are difficult to culture or unculturable, which makes it difficult to observe the dynamic microbial changes throughout the anaerobic digestion process [16]. Advances in next-generation sequencing have resulted in culture-independent tools based on shotgun metagenomics that can potentially profile complex microbial communities [17]. Not only the community structure can be characterized, but the potential functions of certain groups (e.g., methanogenic archaea and acidogenic/hydrolytic bacteria) and their interactions can be identified [18]. While this idea has been explored for centralized municipal wastewater treatment plants (WWTPs) [2,19], there are few studies on the use of high-throughput DNA sequencing for OWTS monitoring where small installations, differing in design and mode of operation, having variable influent (e.g., from single households), are examined.

In this proof of principle study, microbial communities from two types of OWTS anaerobic digesters using conventional box-septic tanks and modified septic tanks equipped with a plug-type digester design plug flow-type were characterized using the whole-metagenome sequencing. The objectives of the study are to survey the microbial consortiums present in conventional and novel digester designs and determine how the microbial community profile and associated functional properties are influenced by digester design and mode of operation.

## 2. Materials and Methods

### 2.1. Digester Site Description

Digester samples were collected from conventional and plug flow-type recirculating and single-pass septic tank installations across Southern and Central Ontario for a total of 12 digesters from September 2018 to January 2019. All systems were located on residential rural properties and served single households. Each OWTS consisted of a digester tank connected to a downstream aerobic biofilter unit. Half of the units were equipped with an aerobic recirculating line (Supplementary Data). The four types of systems were:

- Conventional recirculating septic digesters (CR) ($n = 3$);
- Conventional single-pass septic digesters (CS) ($n = 3$);
- Plug flow-type recirculating septic digesters (PFR) ($n = 3$);
- Plug flow-type single-pass septic digesters (PFS) ($n = 3$).

For each digester, we collected sewage samples at the influent, effluent, and within the tank. In recirculating digesters, an inline valve as part of the recirculating line was used to control the proportion of aerobic effluent being pumped back into the anaerobic septic tanks. Percentages of valve openings for each recirculating digester were listed in Supplementary Data and were set by an industrial partner, Waterloo Biofilter Systems Inc. (Guelph, ON Canada). Hydraulic residence times were calculated by dividing the tank residence volume (L) and flow rate (L/day). Samples were assigned an alphanumeric code, where numbers prefixing the digester acronym represented a digester biological replicate,

and numbers suffixing the digester acronym represented the sampling point, where "1" is for influent, "2" for tank, and "3" for effluent.

### 2.2. Digester Sample Collection

For each digester type and replicate, two effluent, tank, and influent samples were taken for chemical analysis and DNA sequencing. Effluent samples were collected from the spray nozzle feeding the aerobic biofilter unit by placing a collection vessel directly beneath the nozzle. Plug flow influent sewage was collected from the inlet opening of the digester and for tank sewage at the approximate outlet of the plug flow tube within the tank. All chemical measurements except for COD, temperature, pH, and dissolved oxygen were performed by Fleming College's Centre for Advancement of Wastewater Treatment according to protocols described in the CALA Directory of Laboratories listing (member number: 3628) and were based on Standard Methods for the Examination of Water and Wastewater, 22nd ed. [20]. Sewage sample bottles were immediately placed on ice after collection, transported within ice coolers, and stored at −80 °C before DNA extraction. Dissolved oxygen and pH were not measured on site, rather they were performed in the lab after samples were received for processing.

### 2.3. DNA Extraction, Sequencing, and Bioinformatics Analysis

Sewage samples were thawed to room temperature and then vacuum-filtered through sterile 0.22 μm nitrocellulose filters (Millipore, Germany) [21]. Samples were filtered until the membrane filter was completely clogged and no more liquid could pass through [22]. Filters were then folded and fit into sterile capped stock tubes pre-filled with 0.25 μL of 0.1 mm zirconium beads [23] (BioSpec Products, Bartesville, OK, USA). DNA extraction was performed by using the Norgen Biotek Soil DNA Isolation Plus Kit (Norgen Biotek Corp., Thorold, ON, Canada) following the manufacturer's protocol. Extracted DNA samples were quantified using a Qubit Fluorimeter 2.0 (Thermo Fisher Scientific, Waltham, Massachusetts) and stored at −20 °C. The shotgun metagenomic library was prepared by using the NEBNext® Ultra™ II DNA library preparation kit with an insert size of 500 bp. (New England Biolabs Inc., Whitby, Ontario). Paired-end metagenomic sequencing with a read length of 150 bp was performed using HiSeq 2500 Illumina sequencer.

Two FASTQ files (forward and reverse) were obtained for each sample. These FASTQ (quality score "Q score" was equal to or greater than Q30 files processed using Trimmomatic v0.39 [24]. The average reads per sample was 5.3 M reads (SD ± 1.6 M). Quality-filtered sequences were aligned using the DIAMOND-BLASTx algorithm against NCBI's non-redundant (nr) protein database with default BLAST parameters [25]. The BLAST output files were formatted as DIAMOND alignment archives and processed through MEGAN6 V6.18.4 for taxonomic and functional annotation [26].

### 2.4. Statistical Analyses

Count data were normalized using multivariate methods [27]. Relative abundance, diversity indices, rank abundance, rarefaction, and differential abundance graphs were prepared using non-normalized data. All statistical tests were performed using the "phyloseq" R package [28]. Relative abundances graphs were generated by agglomerating (aggregating low-level taxonomic groups) to either the phylum level or, in the case of Euryarchaeota abundances, at the family level. Constrained redundancy analysis (RDA) between microbial community compositions and chemical parameters and non-metric multidimensional scaling ordination methods for all taxonomic and functional samples were generated using the Bray–Curtis dissimilarity index for beta-diversity analysis. PERMANOVA tests across digester types were performed using the vegan package [29].

Comparative SIMPER analysis of Bray–Curtis dissimilarity contributions of taxa was performed using the simper function from the vegan package implemented into custom scripts created by Steinberger named simper.pretty and kruskal.pretty [30]. To identify enriched microbial groups associated with specific digesters, differential abundance pairwise

comparison graphs between digester types (conventional and plug type) and between flow configurations (recirculating and single-pass) were generated using the "DESeq2" package [31,32]. Raw taxonomic counts were assessed using DESeq2 pairwise comparisons [32] of significantly enriched taxa in between digester types and flow configurations. Identified differentially abundant genera were then tested for association with chemical parameters using Spearman rank-order correlations using the R function rcorr in the "Hmisc" package and plotted with the "corrplot" package [33].

## 3. Results and Discussion

### 3.1. Whole-Community Interactions with the Physicochemical Environment

Physicochemical factors such as influent sewage load (COD, TSS), digester design and configuration, and interspecies interactions impact community compositions along the anaerobic digestion process [34]. Intra and inter-digester analysis demonstrated that differences in the microbial community across the four digester types (CR, IR, CS, and IS) were due to dissimilar taxonomic richness across the replicates ($p < 0.05$) [35]. Changes in the bacterial communities between digester groups were also significantly correlated with temperature ($p = 0.029$), $NH_3$ ($p = 0.007$), and pH ($p = 0.003$), suggesting that the physiochemical factors in digesters impacted microbial community composition (Figure 1 and Supplementary Data). Anaerobic digester microbial communities change in response to changes in ammonia/ammonium concentration and pH [36]. Specifically, an excessive accumulation of ammonia/ammonium can lead to inhibition of methanogenesis and favor the dominance of hydrogenotrophic microbial communities [36]. Effects of sampling point, or treatment stages (influent, tank, and effluent), were not significantly different in whole-group ($p > 0.05$) or pairwise comparison tests (Supplementary Data). Therefore, the changes in microbial communities were more likely due to digester designs and physiochemical factors than treatment stages [29,35]. In anaerobic digesters receiving plant, human, or cattle waste feed, the influent composition is one of the major driving forces of anaerobic digestion community specialization [34,37]. Although bacterial community composition in influent samples was not significantly different among the samples, additional replication may be needed to assess the effect of influent composition on the differentiation of downstream communities across digester types (Supplementary Data).

### 3.2. Enriched Microbial Subgroups between Digester Designs

Enrichment of microbial consortium within a community is an indicator of niche functional specializations by specific microbes [38]. Accordingly, methanogenic *Methanothrix* spp. as well as members of Proteobacteria such as *Desulfobulbus* spp., *Desulfomicrobium* spp., *Pelolinea* spp., and *Protecatella* spp. were enriched in conventional systems compared to plug flow-type systems (Figures 1 and 2 and Supplementary Data).

Plug-flow digesters are known to achieve lower COD, higher TSS removal, and methane production rates than other commonly used digester designs such as the conventional single-tube recirculating digester [39]. Consistent with this, we observed lower COD and higher TSS removal in plug-flow digesters than conventional digesters (Figure 3) possibly due to a higher solid retention time resulting in better degradation of solids and untreated sewage by the active microbial biomass (Figure 4) [40]. In addition, hydrogenotrophic methanogens, including Methanospirillaceae, Methanobacteriaceae, and Methanosarcinaceae, were enriched in plug flow-type systems (Supplementary Data). We also observed a positive correlation of TSS with the "Methanogenesis" gene subsystem (Figure 2), possibly due to the higher solid content that was available for methanogens to digest [41].

Sulfate-reducing bacteria such as *Desulfovibrio* spp. were also enriched in plug flow-type systems (Figure 2). The enrichment of hydrogenotrophic methanogens and sulfate-reducing bacterial communities may reflect elevated levels of organic acids since hydrogenotrophs are exclusive $H_2$-scavengers in acidic environments [42].

An inverse relationship was observed between COD, total Kjeldahl nitrogen (TKN), and NH₃ levels with abundances of *Methanothrix* spp. and *Desulfobulbus* spp. (Figure 2) suggesting that these bacterial communities may be associated with biomass degradation in the form of COD decrease and sensitivity to high nitrogen content (in the form of ammonia) [43]. However, lower COD and higher TSS removal in plug flow-type may, in addition to the action of methanogens, also be caused by uncharacterized interspecies interactions. Metagenomic studies have reported rare co-occurring phylotypes that resulted in greater abundances of niche functional features [37,38]. Further investigation may be required to reveal how interspecies and community interactions are responsible for higher waste degradation rates within the plug flow-type than in conventional septic tanks.

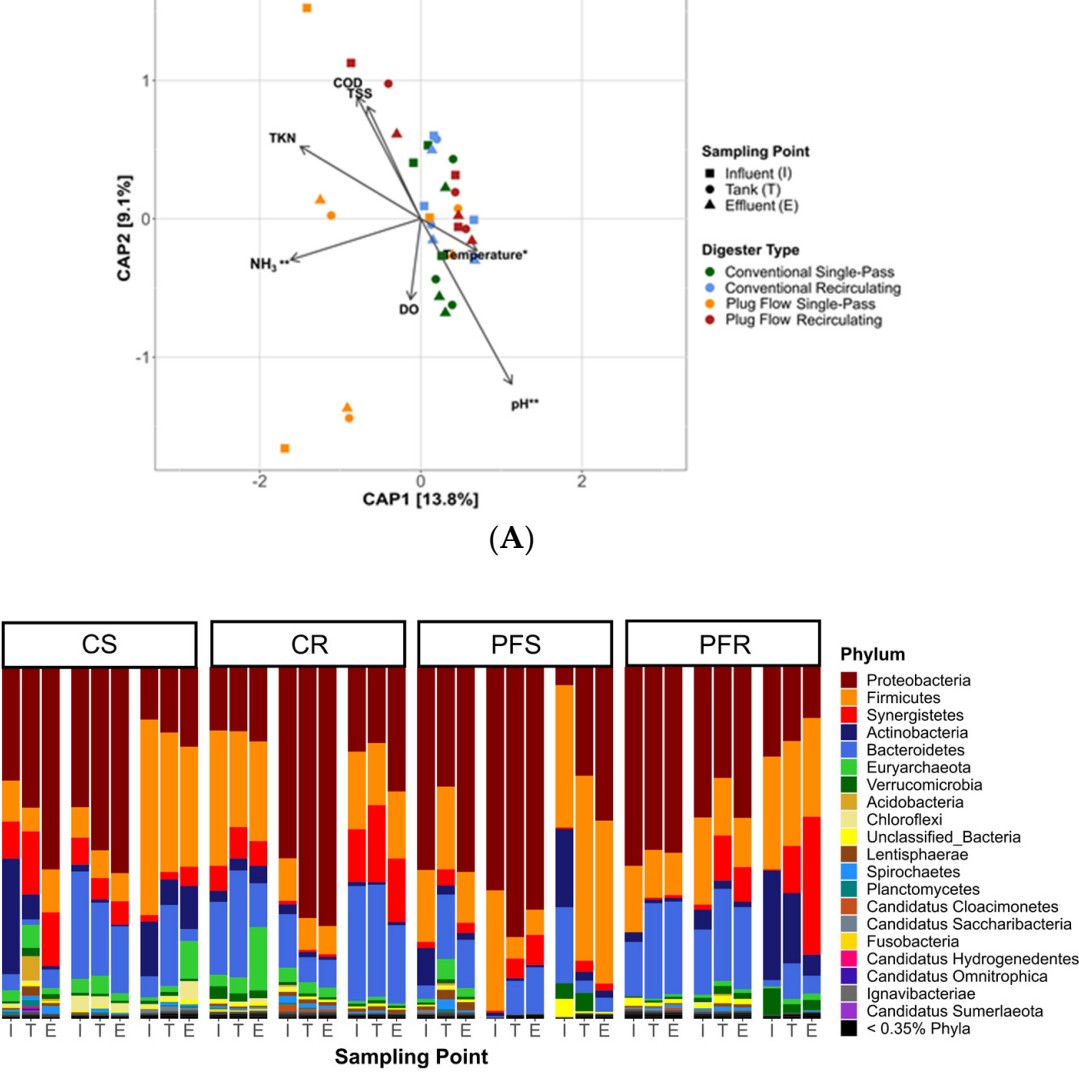

**Figure 1.** (**A**) Redundancy analysis of digester type and sampling point constrained against chemical parameters. Chemical parameters that significantly influenced the ordination revealed by the PERMANOVA test are denoted with asterisks (adj. *p* < 0.05), where "**" 0.01, "*" 0.05. (**B**) Relative abundance bar graphs showing the top 20 phyla for triplicate plug flow-type single-pass (PFS), plug flow-type recirculating (PFR), conventional single-pass (CS), and conventional recirculating (CR) across sampling points. The lowest abundant phyla were aggregated into a cutoff percentage labeled in black.

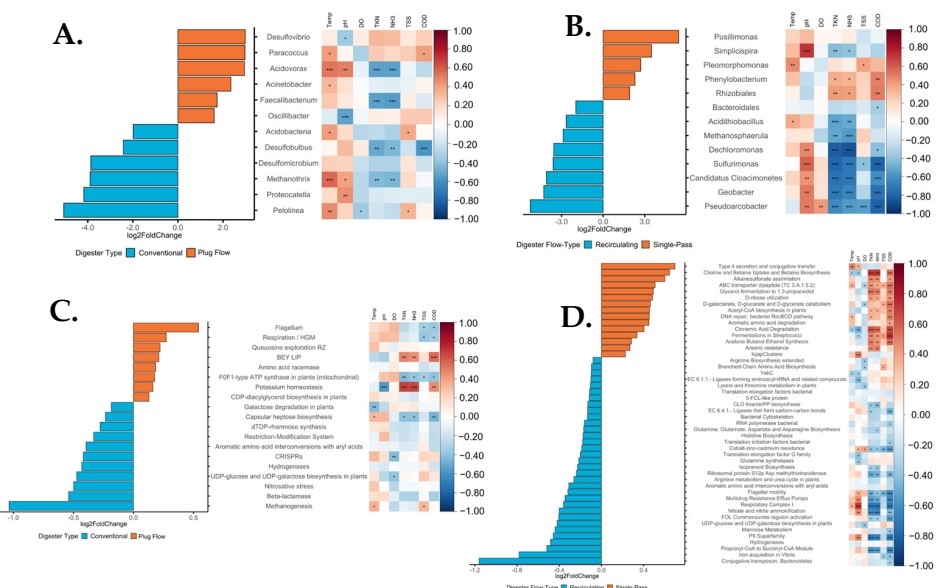

**Figure 2.** Pairwise comparisons with Spearman rank correlations to chemical parameters of differentially abundant genera in (**A**) conventional vs. plug flow-type, (**B**) recirculating and single-pass and differentially abundant functional genes in (**C**) conventional vs. plug flow-type, (**D**) recirculating and single-pass. Differential abundance analysis was performed using DESeq2 with an abundance cutoff of 0.1% of total agglomerated genera and subsystem Level III counts and an adjusted *p*-value of $p < 0.001$ for genera and $p < 0.05$ for gene counts. Significant Spearman correlations are marked with asterisks (adj. $p < 0.05$), where "***" 0.001, "**" 0.01, "*" 0.05.

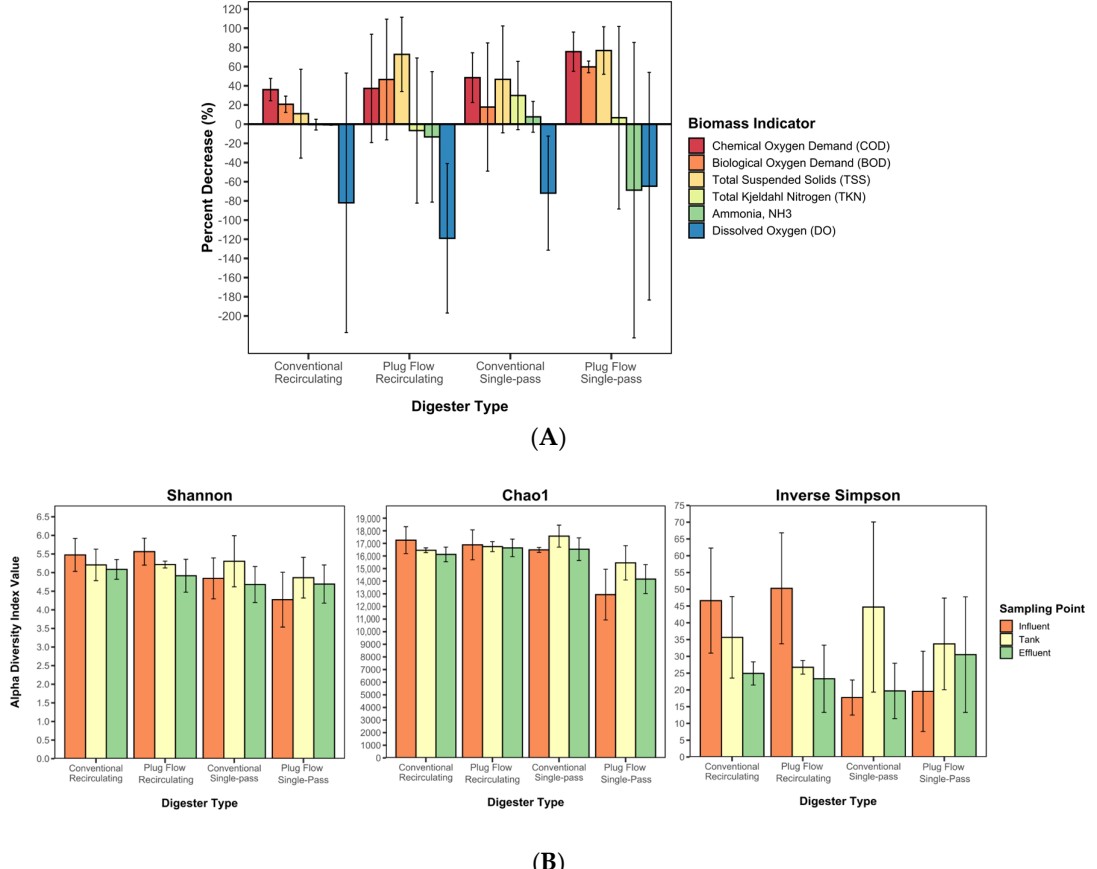

**Figure 3.** (**A**) Mean percent decrease in biomass chemical parameters across influent, tank, and

effluent sampling points. Error bars indicate standard deviation. (**B**) Mean Shannon diversity, Chao1 richness, and inverse Simpson evenness indices of the four digester types over influent, tank, and effluent sewage. Error bars indicate standard error (*n* = 3).

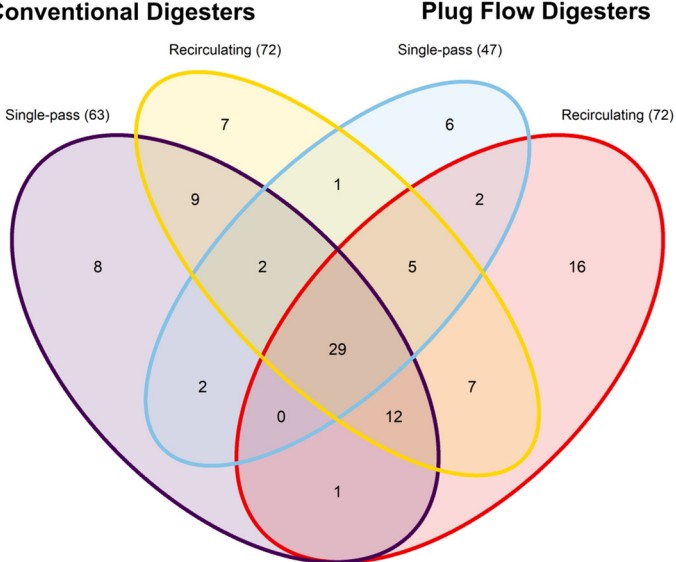

**Figure 4.** Shared and exclusive genera that are at least 50% prevalent across conventional single-pass (CS), conventional recirculating (CR), plug flow-type single-pass (PFS), and plug flow-type recirculating (PFR) and digesters (see Supplementary Data for list of taxa).

### 3.3. Enriched Microbial Subgroups among Digester Flow Configurations

Implementing downstream aerobic recirculating lines in anaerobic systems facilitates the anaerobic digestion process by replenishing nitrate for consumption by nitrate-reducing microbes (i.e., denitrifiers), preventing ammonia accumulation, and providing acidic buffering capacity by ensuring the constant use and flow of $H_2$ by denitrifiers [44]. Niche functional specialization was observed in recirculating systems, with recirculated digesters harboring the highest number of exclusive genera and enriched with microbial groups that were negatively correlated with TKN, $NH_3$, TSS levels, and COD (Figures 2 and 4 and Supplementary Data). The majority of these have roles in sulfur metabolisms and denitrification facilitated by aerobic effluent recirculation [45] (Figures 1 and 2 and Supplementary Data). The negative correlations between the recirculating-enriched bacteria and the biomass indicators (TKN, $NH_3$, TSS, and COD) suggested that these bacterial communities were involved in waste degradation.

Sulfur-metabolizing bacteria such as *Pseudoarcobacter*, *Sulfurimonas*, and *Geobacter* were enriched in a recirculating system. *Pseudoarcobacter* and *Sulfurimonas* are anaerobic, sulfur-oxidizing bacteria coupling denitrifying processes [46,47], while *Geobacter* spp. are sulfur-reducers, reducing elemental sulfur to sulfide [45]. The enrichment of sulfur-oxidizing denitrifiers in recirculating systems suggested higher nitrogen and propionate metabolization potential in recirculating systems, also observed from the elevated abundances of functional genes related to nitrogen metabolism and propionate oxidization (Figure 2).

In single-pass systems, compared to recirculating, sulfur-oxidizing denitrifiers and hydrogenotrophic methanogens were the main bacterial groups involved in anaerobic digestion. These results support the idea that septic tank design and flow configuration can impact the growth of specific microbial groups that facilitate nutrient removal in on-site wastewater treatment systems [47].

### 3.4. The Succession of Microbial Subgroups in OWTS Anaerobic Digesters

Since anaerobic digestion is a stepwise process [39,48], we focused on the succession of four functional gene subgroups in hydrolytic, acidogenic, acetogenic, and methanogenic

processes in anaerobic digesters. From influent to tank to effluent, increasing relative abundances of genes in methanogenic, acidogenic, and acetogenic subgroups (glutamate, glutamine, aspartate, and asparagine metabolism under the category of amino acid and derivatives) in both conventional and plug flow-type recirculating systems (Figures 5 and 6), likely a reflection of progressive maturation of the anaerobic digestion community. Conversely, the relative abundance of hydrolytic gene subgroup (lactose and galactose uptake and use under the category of carbohydrate metabolism) decreased.

Influent sewage contained higher abundances of putative gut bacteria Lachnospiraceae and Ruminococcaceae, which were likely derived from the human sewage feed. These bacteria possess hydrolytic and acidogenic functions [39]. Effluent sulfur-reducing bacteria, nitrifying bacteria, and methanogens were significantly correlated with biomass indicators (Figure 5). Effluent-enriched sulfate-reducing deltaproteobacteria included *Desulfamplus* spp. [49], *Desulfovibrio* spp. [47], *Desulfobacterium* spp. and *Desulfobacter* spp. [50]. Elemental sulfur-reducing bacteria, including *Geobacter* spp. [45], *Desulfonatronum* spp. [51], *Desulfuromonas* spp. [52] and *Desulfobulbus* spp. [50] were also enriched in the effluent. *Methanomethylvorans* spp. are obligately methylotrophic methanogens using methanol as an electron donor [53], were also enriched in the effluent. In recirculating digesters (both conventional and plug flow-type), the alpha diversity from influent to effluent decreased slightly (Figure 3). In contrast, single-pass digesters (both conventional and plug flow-type) had the highest alpha diversity within the tank, followed by the effluent and influent (Figure 3). Effluent of recirculating digesters had higher diversity than the influent was likely due to enrichment of denitrifiers and hydrogenotrophic methanogens, in response to elevated nitrate levels replenished by the recirculating line [48].

Hydrolytic and fermentative bacteria accumulate at the early stages of anaerobic digestion, which convert complex polysaccharides, proteins, and long-chain fatty acids to volatile fatty acids, propionate, and butyrate [39,48]. At late stages, acetogens oxidize volatile and short-chain fatty acids to produce $H_2$ so that methanogens can use it to produce methane. Consistent with this stratification, we found gut bacteria in the influent and sulfidogenic and methanogenic microbes was enriched in the effluent. Ultimately, this may have resulted in the observed decrease in COD and TKN (Figure 5). Most of the enriched genera in effluent, such as the sulfidogenic and methanogenic microbes, were at low abundances (<5%). However, bacterial communities with lower abundance may still influence anaerobic digester functions [38]. Similarly, our results demonstrated that genes related to nitrogen metabolization, sulfur reduction, and methanogenic pathways were associated with bacteria that are in low abundance.

While this study indicates that DNA sequencing can be a useful additional tool for OWTS monitoring, additional studies are needed to identify how much replication, beyond the $n = 3$ used in this study, is needed by using power analyses to provide satisfactory information to operators and, potentially, to regulatory agencies that certify OWTS technology (Supplementary Data). In addition, as many parameters influence anaerobic digester performance, evaluating additional endpoints that are not routinely monitored by operators, particularly $CO_2$ and methanogenesis, may be desirable. This may, for example, establish the degree of association between biogas production and specific microbial community composition.

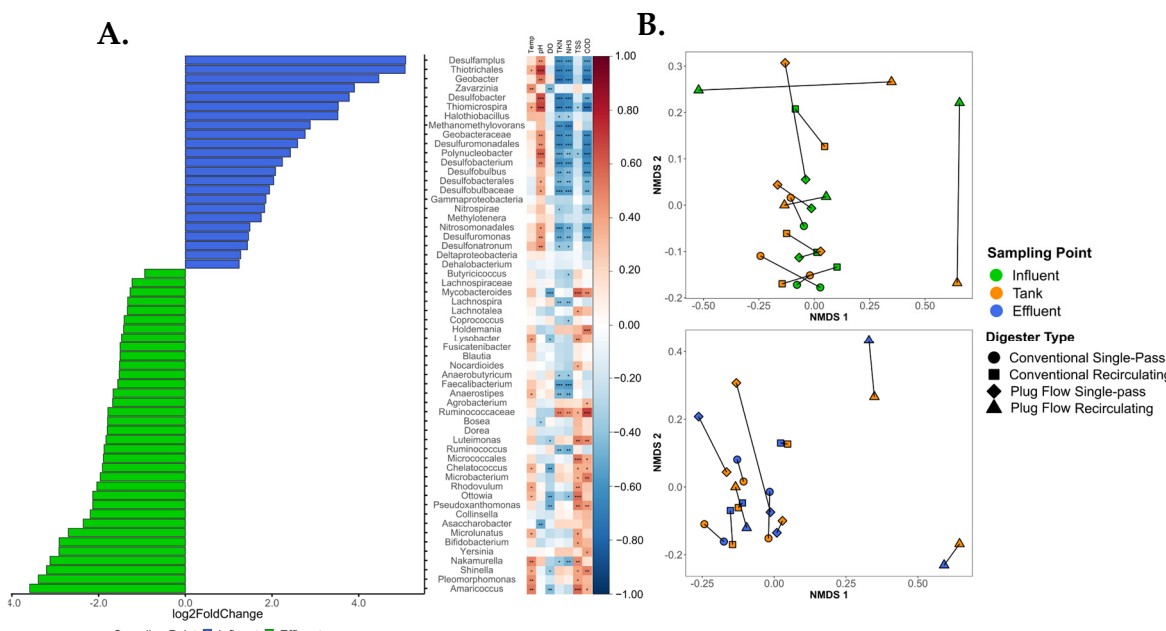

**Figure 5.** (**A**) Pairwise comparisons of differentially abundant genera between and Spearman rank correlations of influent and effluent sewage. Differential abundance analysis was determined using DESeq2 with an abundance cutoff at 0.01% of total agglomerated genera counts and an adjusted *p*-value cutoff at $p < 0.05$. Significant correlations are marked with asterisks ($p < 0.05$), where "***" 0.001, "**" 0.01, "*" 0.05. (**B**) Procrustes NMDS ordination influent vs. tank and tank vs. effluent comparisons. Significance test of the M2 procrustes statistic was performed by the R vegan function "protest" ($p < 0.05$).

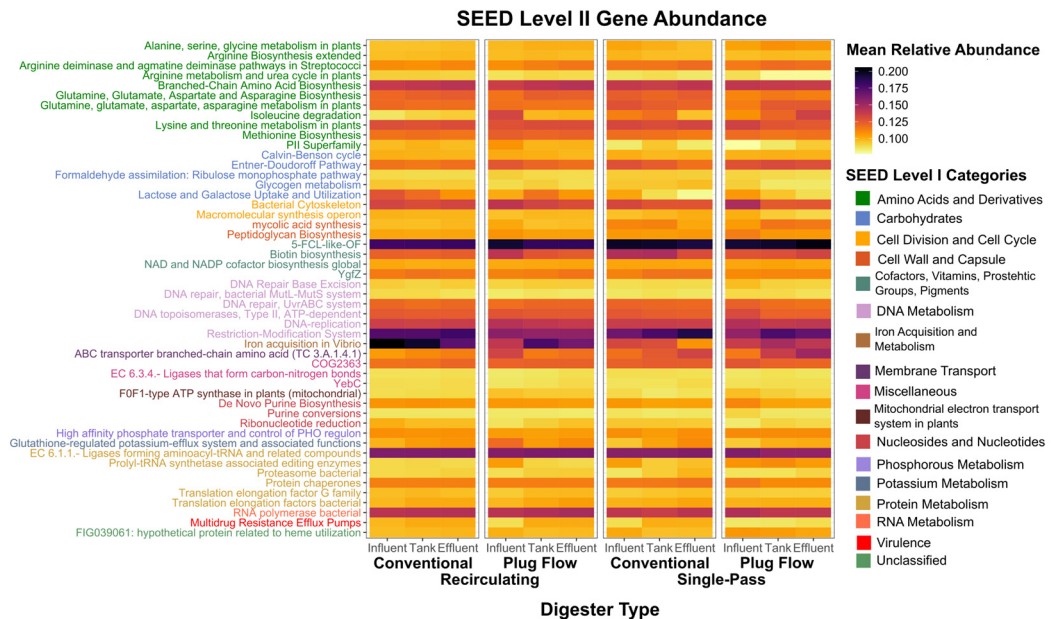

**Figure 6.** Top 50 most abundant functional gene subsystems (SEED Level III). Level III gene subsystems categorized alphabetically by Level I classifications are indicated in color.

## 4. Conclusions

- Plug flow-type anaerobic digesters exhibited a comparatively higher abundance of hydrogenotrophic methanogens and sulfate-reducing bacterial communities, while conventional digesters harbored greater propionate-oxidizing acetogen and acetoclastic methanogen populations;
- Sulfur-metabolizing bacterial communities were found to be enriched in recirculating systems, which was also supported by the increased number of nitrogen and propionate metabolization genes;
- Sulfur-oxidizing denitrifiers and hydrogenotrophic methanogens were the most abundant anaerobic digestor bacterial communities relative to recirculating systems;
- Although hydrogenotrophic methanogens and sulfur-reducing bacterial communities represented <5% of the total bacterial population, less dominant taxonomic groups may still influence anaerobic digestion processes;
- Next-generation sequencing tools represent a promising new tool, augmenting traditional technology, for monitoring anaerobic digesters in on-site waste treatment systems.

**Supplementary Materials:** Supporting information can be found in the electronic version of this article at https://www.mdpi.com/article/10.3390/pr10030436/s1.

**Author Contributions:** Conceptualization, H.E.S.; methodology, H.E.S., J.N. and A.W.Y.C.; validation, H.E.S., F.S. and E.L.; formal analysis, J.N.; writing—original draft preparation, J.N.; writing—review and editing, H.E.S., F.S., E.L. and J.D.; supervision, H.E.S. All authors have read and agreed to the published version of the manuscript.

**Funding:** This study was supported with funding from the NSERC Engage program (492465-15), the Ontario Water Consortium (SUB02625), and the Ontario Research Excellence fund (RE09-77). We thank Craig and Chris Jowett of Waterloo Biofilter Systems for technical advice. We also thank Waterloo Biofilter for their in-kind support of this project.

**Institutional Review Board Statement:** Not applicable.

**Informed Consent Statement:** Not applicable.

**Data Availability Statement:** Data will be available upon request.

**Acknowledgments:** We thank members of the Schellhorn lab for helpful discussions and comments on the manuscript.

**Conflicts of Interest:** The authors declare no conflict of interest.

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
