# Peer review of "Comparative Metagenomics of Anaerobic Digester Communities Reveals Sulfidogenic and Methanogenic Microbial Subgroups in Conventional and Plug Flow Residential Septic Tank Systems"

_processes, doi:10.3390/pr10030436_

Round 1
Reviewer 1 Report
Manuscript ID processes-1601045 entitled “Comparative Metagenomics of Anaerobic Digester Communities Reveals Sulfidogenic and Methanogenic Microbial Subgroups in Conventional and Plug-Flow Residential Septic Tank Systems” is an interesting and up-to-date look at the functioning of anaerobic reactors through the prism of the type of anaerobic bacterial groups present in the population.
The analysis and monitoring of the taxonomic structure and knowledge of the profile of the bacterial community can be a valuable supplement to technological information on the functioning of anaerobic wastewater treatment systems.
In my opinion, the subject of the manuscript fully corresponds to the profile of the Processes journal. The authors correctly presented the methodological assumptions of the works and the obtained results. They applied appropriate statistical tests. In my opinion, the manuscript requires a few additions and corrections. My remarks below:
- A graphic abstract would be a valuable supplement to the content of the manuscript, which would help the reader understand the main purpose of the research work being carried out.
- Keywords should be supplemented with: metagenomics, bacterial communities.
- The introduction is very laconic and needs to be developed. It seems that the basic technological parameters of the process, such as HRT, OLR, temperature or the type of anaerobic reactor, may have (as proved by the authors) the characteristics of the bacterial community. It seems, however, that the method of monitoring the operation of anaerobic reactors proposed by the authors may have wider applications when the anaerobic reactors are supported by other factors that may significantly modify the taxonomic structure of anaerobic bacteria.
- In modern solutions and designs of reactors, the fermentation process is often supported by physical factors, which significantly modify the taxonomic structure of anerobes. There, the method proposed by the authors can find wider and common applications. Introduction and discussion sections should be supplemented with this kind of information: https://doi.org/10.3390/en14030590, https://doi.org/10.1016/j.jclepro.2020.122664, https://doi.org/10.1007/s11356 -021-14095-y, https://doi.org/10.1016/j.biortech.2015.10.037, https://doi.org/10.1007/s00253-016-7321-2, https://doi.org /10.3390/pr9101772 and many others.
- Please correct / remove the recurring error in the text resulting from the automatic assignment of cited items (Error! Reference source not found).
- In the technology of anaerobic degradation of pollutants, a very important factor, apart from the efficiency of wastewater treatment, is the efficiency of biogas production and the content of methane.
- A very valuable information and supplement to the manuscript would be the presentation and an attempt to explain the relationship between the bacterial community and the efficiency of the production of gaseous changes in anaerobic bacteria.
- In my opinion, this is an interesting manuscript. Good luck !!!
Reviewer 2 Report
Abstract
- Line 14 and 21: Avoid active sentence. Remove ‘we and rephrase the sentence.
- COD, TSS, DNA: Abbreviations should be defined at first mention in the main text.
Introduction:
- General: It is suggested to emphasize on advantage of monitoring the performance using this technique and include the limitation of physicochemical approach.
- Line 47-49 : citation needed
Materials and research method
2.3: This section require revision. kindly rephrase the sentence, too long and complex. This sentence is a little bit heavy, please reformulate
Result and Discussion
1.Overall, all figures presented in this section are not clear enough.
2.Error in citation
- It is suggested for the authors to classify indicators individually. For example, which result are most related and can represented TSS and BOD respectively.
Round 2
Reviewer 1 Report
The manuscript has been revised in line with my suggestions and comments. In my opinion, this is an interesting and valuable research material and can be published in its current form.